# Better Best of Both Worlds Bounds
# for Bandits with Switching Costs

**Idan Amir**
Department of EE
Tel Aviv University
idanamir@mail.tau.ac.il

**Guy Azov**
Department of EE
Tel Aviv University
guyazov@mail.tau.ac.il

**Tomer Koren**
Blavatnik School of CS, Tel Aviv University
and Google Research
tkoren@tauex.tau.ac.il

**Roi Livni**
Department of EE
Tel Aviv University
rlivni@tauex.tau.ac.il

## Abstract

We study best-of-both-worlds algorithms for bandits with switching cost, recently addressed by Rouyer, Seldin, and Cesa-Bianchi [14]. We introduce a surprisingly simple and effective algorithm that simultaneously achieves minimax optimal regret bound (up to logarithmic factors) of $\mathcal{O}(T^{2/3})$ in the oblivious adversarial setting and a bound of $\mathcal{O}(\min\{\log(T)/\Delta^2, T^{2/3}\})$ in the stochastically-constrained regime, both with (unit) switching costs, where $\Delta$ is the gap between the arms. In the stochastically constrained case, our bound improves over previous results due to [14], that achieved regret of $\mathcal{O}(T^{1/3}/\Delta)$. We accompany our results with a lower bound showing that, in general, $\tilde{\Omega}(\min\{1/\Delta^2, T^{2/3}\})$ switching cost regret is unavoidable in the stochastically-constrained case for algorithms with $\mathcal{O}(T^{2/3})$ worst-case switching cost regret.

## 1 Introduction

Multi Armed Bandit (MAB) is one of the most fundamental problems in online learning and sequential decision making. This problem is often framed as a sequential game between a player and an environment played over $T$ rounds. In each round, the player chooses an action from a finite set $[K] = \{1, \ldots, K\}$ and incurs a loss in $[0, 1]$ for that action. The environment then only reveals the loss of the chosen action—this is referred to as bandit feedback. The goal of the player is to minimize the *regret*, which measures the difference between the cumulative loss of the player and that of the best arm in hindsight.

Two common regimes often studied in the MAB literature are the *adversarial* setting [6] and the so-called *stochastically-constrained* setting [18] which is a generalization of the more classical stochastic setting. In the former regime, losses are generated arbitrarily and possibly by an adversary; in the latter, losses are assumed to be generated in a way that one arm performs better (in expectation) than any other arm, by a margin of $\Delta > 0$. Both regimes have witnessed a flurry of research [1, 3, 5, 6, 12, 17] leading to optimal regret bounds in each of the settings.

Recently, significant effort has been dedicated for designing *best-of-both-worlds* MAB algorithms, where one does not have a-priori knowledge on the underlying environment but still wishes to enjoy the optimal regret in both regimes simultaneously [4, 7, 15, 16, 18, 19]. Most notably, Zimmert and Seldin [19] analyzed the *Tsallis-INF* algorithm and established that it achieves optimal regret bounds

in both stochastically-constrained and adversarial environments, matching the corresponding lower bounds asymptotically.

Another well-studied variant of the MAB setting is that of Bandits with switching cost [2, 8–10], where the learner suffers not only regret but also a penalty for switching actions. As shown by Dekel et al. [8] adding a unit switching cost to the regret incurs a lower bound of $\tilde{\Omega}\left(K^{1/3}T^{2/3}\right)$, in contrast to $\Omega(\sqrt{KT})$ in the standard setting, highlighting the difficulty of this setup. Recently Rouyer et al. [14] asked the question of how best-of-both-worlds algorithms can be obtained for MAB where switching costs are considered. Rouyer et al. [14] managed to show a best-of-both-worlds type algorithm that, for constant $K$ and cost-per-switch, achieves optimal regret bound of $\mathcal{O}(T^{2/3})$ in the oblivious adversarial regime, whereas in the stochastically-constrained setup their upper bound is of order $\mathcal{O}(T^{1/3}/\Delta)$, which was unknown to be optimal.

In this work we tighten the above gap. we introduce a new algorithm that improves the bound of [14] and achieves $\mathcal{O}(\min\{\log T/\Delta^2, T^{2/3}\})$ in the stochastically constrained case (while obtaining the same optimal regret bound in the worst-case). Further, we provide a lower regret bound, and show that the above bound is tight, up to logarithmic factors, in the best-of-both-worlds setup. For the more general case of $K > 2$ arms, our algorithm still improves over [14], however in that case our lower bounds and upper bound do not fully coincide; we leave this as an open question for future study.

## 2 Setup and Background

In the classic Multi Armed Bandit problem with $K$ arms, a game is played consecutively over $T > K$ rounds. At each round $t \le T$, an adversary (also called the environment) generates a loss vector $\ell_t \in [0, 1]^K$. The player (referred to as learner) selects an arm $I_t \in [K]$ according to some distribution $p_t \in \Delta^K$ where $\Delta^K := \{p \in [0, 1]^K : \sum_{i \in [K]} p_i = 1\}$, and observes $\ell_{t,I_t}$, which is also defined to be its loss at round $t$. Notice that the learner never has access to the entire loss vector $\ell_t \in [0, 1]^K$.

The performance of the learner is measured in terms of the *regret*. The regret of the learner is defined as

$$\mathcal{R}_T := \sum_{t \in [T]} \ell_{t,I_t} - \min_{i \in [K]} \sum_{t \in [T]} \ell_{t,i}.$$

Another common performance measure we care about is the *pseudo-regret* of the algorithm:

$$\overline{\mathcal{R}}_T := \sum_{t \in [T]} \ell_{t,I_t} - \min_{i \in [K]} \sum_{t \in [T]} \mathbb{E}\left[\ell_{t,i}\right].$$

We next describe two common variants of the problem, which differ in the way the losses are generated.

**Adversarial (oblivious) regime:** In the *oblivious adversarial* regime, at the beginning of the game the environment chooses the loss vectors $\ell_1, \ldots, \ell_T$, and they may be entirely arbitrary. In general, the objective of the learner is to minimize its expected regret for the adversarial regime. One can observe that the *expected pseudo-regret* coincides with the *expected regret*, in this setting. More generally, it can be seen that the expected regret upper bounds the expected pseudo-regret. Namely, $\mathbb{E}[\overline{\mathcal{R}}_T] \le \mathbb{E}[\mathcal{R}_T]$.

**Stochastically-constrained adversarial regime:** We also consider the *stochastically-constrained adversarial* regime [18]. In this setting we assume that the loss vectors are drawn from distributions such that there exists some $i^\star \in [K]$,

$$\forall i \ne i^\star : \ \mathbb{E}[\ell_{t,i} - \ell_{t,i^\star}] = \Delta_i, \tag{1}$$

independently of $t$[1]. That is, the gap between arms remains constant throughout the game, while the losses $\{\ell_{t,i}\}_{t \in [T]}$ of any arm $i$ are drawn from distributions that are allowed to change over time

---

[1]This definition is equivalent to the standard definition of $\forall i, j : \ \mathbb{E}[\ell_{t,i} - \ell_{t,j}] = \Delta_{i,j}$.

and may depend on the learner's past actions $I_1, \ldots, I_{t-1}$. It is well known that the stochastically-constrained adversarial regime generalizes the well-studied stochastic regime that assumes the losses are generated in an i.i.d. manner.

We denote the best arm at round $t$ to be $i_t^\star = \arg\min_{i \in [K]} \mathbb{E}[\ell_{t,i}]$. Note that since the gap between arms is constant we have that $\forall t \in [T] : i_t^\star = i^\star$ where $i^\star$ is the optimal arm. We consider the case where there is a *unique* best arm. Also, we denote the gap between arm $i$ and $i^\star$ to be $\Delta_i := \mathbb{E}[\ell_{t,i} - \ell_{t,i^\star}]$ and we let $\Delta_{\min} = \min_{i \neq i^\star} \Delta_i$.

We note that in the stochastically-constrained case, the pseudo-regret is often expressed by the sub-optimality gaps $\Delta_i$, and it is given by:

$$\mathbb{E}[\overline{\mathcal{R}}_T] := \sum_{t \in [T]} \sum_{i \neq i^\star} \mathbb{P}(I_t = i) \Delta_i.$$

## 2.1 Multi-Armed Bandits with Switching Cost

In the problem described above, there is no limitation on the number of times the player is allowed to switch arms between consecutive rounds. In this work, we consider a setup where the regret is accompanied by a switching cost, as suggested by Arora et al. [2]. We then measure our performance by the switching cost regret, parameterized by the switching cost parameter $\lambda \geq 0$ :

$$\overline{\mathcal{R}}_T^\lambda := \overline{\mathcal{R}}_T + \lambda \mathcal{S}_T,$$

where $\mathcal{S}_T := \sum_{t \in [T]} \mathbb{1}\{I_t \neq I_{t-1}\}$.

**Best-of-both-worlds with switching cost.** Rouyer et al. [14] considered the setting of switching cost in the framework of best-of-both-worlds analysis. They showed (Thm 1 therein): that there exists an algorithm, Tsallis-Switch, for which in the adversarial regime the pseudo-regret of Tsallis-Switch for $\lambda \in \Omega(\sqrt{K/T})$ :

$$\mathbb{E}[\overline{\mathcal{R}}_T^\lambda] \leq \mathcal{O}\left((\lambda K)^{1/3} T^{2/3}\right), \tag{2}$$

and in the stochastically constrained setting:

$$\mathbb{E}[\overline{\mathcal{R}}_T^\lambda] \leq \mathcal{O}\left(\sum_{i \neq i^\star} \frac{(\lambda K)^{2/3} T^{1/3} + \log T}{\Delta_i}\right). \tag{3}$$

# 3 Main results

Our main result improves over the work of Rouyer et al. [14] and provides an improved best-of-both-worlds algorithm for the setting of switching cost

**Theorem 1.** *Provided that $\lambda \geq \sqrt{K/T}$, the expected pseudo-regret with switching cost of "Switch Tsallis, Switch!" (Algorithm 1) satisfies the following simultaneously:*

- *In the adversarial regime,*

$$\mathbb{E}[\overline{\mathcal{R}}_T^\lambda] = \mathcal{O}((\lambda K)^{1/3} T^{2/3}). \tag{4}$$

- *In the stochastically constrained adversarial regime,*

$$\mathbb{E}[\overline{\mathcal{R}}_T^\lambda] = \mathcal{O}\left(\min\left\{\left(\frac{\lambda \log T}{\Delta_{min}} + \log T\right) \sum_{i \neq i^\star} \frac{1}{\Delta_i}, (\lambda K)^{1/3} T^{2/3}\right\}\right). \tag{5}$$

We next compare the bound of "Switch Tsallis, Switch!" and Rouyer et al. [14]. We first observe that for small switching cost, $\lambda \leq O(\sqrt{K/T})$, both algorithms basically ignore the switching cost and run standard Tsallis without any type of change hence the algorithms actually coincide, so we only care for the case $\lambda \geq \sqrt{\frac{K}{T}}$. Also, notice that in the adversarial regime Eq. (2) and Eq. (4) are equivalent and both algorithms obtain the minimax optimal regret (up to logarithmic factors). In the stochastically

constrained regime comparing Eqs. (3) and (5), note that for $\Delta_{\min} \le (\lambda K)^{1/3} T^{-1/3} \log T$, we have that

$$\sum_{i \ne i^\star} \frac{(\lambda K)^{2/3} T^{1/3}}{\Delta_i} \ge \frac{(\lambda K)^{2/3} T^{1/3}}{\Delta_{\min}} = \tilde\Omega \left( (\lambda K)^{1/3} T^{2/3} \right),$$

which is comparable to our bound up to logarithmic factors. On the other hand, if $\Delta_{\min} \ge (\lambda K)^{1/3} T^{-1/3} \log T$

$$\sum_{i \ne i^\star} \frac{(\lambda K)^{2/3} T^{1/3}}{\Delta_i} \ge \left( \frac{\lambda K \log T}{\Delta_{\min}} \right) \sum_{i \ne i^\star} \frac{1}{\Delta_i} = \Omega \left( \frac{\lambda \log T}{\Delta_{\min}} \sum_{i \ne i^\star} \frac{1}{\Delta_i} \right),$$

which is comparable to Eq. (5). It can also be observed that when $\Delta_{\min}$ is large enough (larger than say $T^{-1/3} \log T$) our bound improves over Eq. (3) by a factor of $\tilde{\mho}(T^{1/3} \Delta_{\min})$.

Next, we describe a lower bound, which demonstrates that our bounds are tight for $K = 2$ (up to logarithmic factors).

**Theorem 2.** *Let A be a randomized player in a multi armed bandit game of K arms played over T rounds with a switching cost regret guarantee of $\mho(K^{1/3} T^{2/3})$ in the adversarial regime. Then, for every $\Delta > 0$ there exists a stochastically-constrained sequence of losses $\ell_1, \ldots, \ell_T$ with minimal gap parameter $\Delta$, that A incurs $\overline{\mathcal{R}}_T + \mathcal{S}_T = \tilde\Omega\big(\min\{1/\Delta^2, K^{1/3} T^{2/3}\}\big)$.*

## 4   Algorithm

Our algorithm, "Switch Tsallis, Switch!" (see Algorithm 1), is a simple modification of Tsallis-INF. We start by playing the original Tsallis-INF algorithm introduced by [19], and after a certain amount of switches we switch to a second phase that plays a standard block no-regret algorithm.

---

**Algorithm 1** Switch Tsallis, Switch!

---

    **Input:** time horizon $T$, switching cost $\lambda$.
1: Initialize: $S = 0$, $\hat\ell_0 = \mathbf{0}_K$, $\eta_t = 2/\sqrt{t}$
2: **for** $t = 1, \ldots, T$ **do**    % Run standard Tsallis Inf
3:     Compute:

$$p_t = \arg\min_{p \in \Delta^K} \left\{ \sum_{r=0}^{t-1} \hat\ell_r \cdot p - \frac{1}{\eta_t} \sum_{i \in [K]} 4\sqrt{p_i} \right\}.$$

4:     Sample $I_t \sim p_t$, play $I_t$ and observe the loss $\ell_{t,I_t}$.
5:     Update:

$$\forall i \in [K]: \ \hat\ell_{t,i} = \frac{\ell_{t,i}}{p_{t,i}} \mathbb{1}\{I_t = i\}$$

$$S = S + \mathbb{1}\{I_t \ne I_{t-1}\}$$

6:     **if** $S \ge K^{1/3} (\frac{T}{\lambda})^{2/3}$ **then**
7:         **break**
8: **if** $t < T$ **then** for remaining rounds
9:     Run Tsallis-INF over blocks (Algorithm 2) of size $\lambda^{2/3} K^{-1/3} T^{1/3}$.

---

The idea is motivated by our observation that under the stochastically constrained setting, there is a natural bound on the number of switches which is of order $O(\overline{\mathcal{R}}_T / \Delta_{\min})$, so as long as this number doesn't exceed the worst case bound of the adversarial setting we have no reason to perform blocks. In other words, we start by playing under the assumption that we are in the stochastically-constrained regime and if the number of switches is larger than expected, we break and move to an algorithm that handles only the oblivious adversarial case.

Best-of-both-worlds algorithms that start under stochasticity assumption and break are natural to consider. Indeed, in the standard setting, without switching they were studied and suggested [4, 7]. However, while successful at the stochastic case, they suffer from a logarithmic factor in the adversarial regime. Moreover, in the standard best-of-both-worlds setup (without switching cost),

the optimal methods don't attempt to identify the regime (stochastic or adversarial). In contrast, what we observe here, is that once switching cost is involved, the criteria to shift between the regimes becomes quite straightforward which allows us to design such a simple algorithm. Indeed, unlike regret which is hard to estimate, the cost of switching is apparent to the learner, hence we can verify when our switching loss exceeds what we expect in the adversarial regime and decide to switch to enjoy both worlds.

# 5    Proofs

Before delving into the proof, we provide a brief review of the Tsalis-INF algorithm guarantees , introduced by Zimmert and Seldin [19], which serves as the backbone of Algorithm 1.

## 5.1    Technical Preliminaries

**Theorem 3** ([19, Thm 1]). *The expected pseudo-regret of Tsallis-INF under the adversarial regime satisfies:*

$$\mathbb{E}[\overline{\mathcal{R}}_T] \leq 4\sqrt{KT} + 1, \tag{6}$$

*and in the stochastically constrained setting:*

$$\mathbb{E}[\overline{\mathcal{R}}_T] \leq \mathcal{O}\left(\sum_{i \neq i^\star} \frac{\log T}{\Delta_i}\right) \tag{7}$$

---

**Algorithm 2** Mini-Batched Tsallis-INF

---

**Input:** time horizon $T$, block size $B$.
1: Initialize: $\hat{\ell}_0 = \mathbf{0}_K$, $\eta_n = 2/\sqrt{n}$, $|B_n| = B$
2: **for** $n = 1, ..., T/B$ **do**
3:       Compute:

$$p_n = \arg\min_{p \in \Delta^K} \left\{ \sum_{r=0}^{n-1} \hat{\ell}_r \cdot p - \frac{1}{\eta_n} \sum_{i \in [K]} 4\sqrt{p_i} \right\}.$$

4:       Sample $I_n \sim p_n$ and play $I_n$ for $B$ times.
5:       Suffer the loss $\sum_{t \in B_n} \ell_{t,I_n}$ and observe its average: $\frac{1}{B}\sum_{t \in B_n} \ell_{t,I_n}$
6:       Update:

$$\forall i \in [K]: \ \hat{\ell}_{n,i} = \frac{\frac{1}{B}\sum_{t \in B_n} \ell_{t,I_n}}{p_{n,i}} \mathbb{1}\{I_n = i\}$$

---

Arora et al. [2] showed how, given a no-regret algorithm against an oblivious adversary, one can convert the algorithm to be played over mini-batches of size $B$ and obtain a regret of $B \, \mathbb{E}\left[\mathcal{R}_{T/B}\right]$. In turn, the regret with switching cost is bounded by $\mathbb{E}\left[\mathcal{R}_T^\lambda\right] \leq B \, \mathbb{E}\left[\mathcal{R}_{T/B}\right] + \lambda\frac{T}{B}$. Applying their approach to build a mini-batched version of Tsallis-INF leads to Algorithm 2 which we depict above, yielding a similar algorithm to a one suggested by Rouyer et al. [14]. Using the above bound for the special case of Tsallis-INF, we obtain the following guarantee:

**Corollary 4.** *There exists an algorithm ,in particular - Algorithm 2 with constant blocks of size $B = \mathcal{O}\left(\lambda^{2/3}K^{-1/3}T^{1/3}\right)$ , for which the expected regret with switching cost, satisfies*

$$\mathbb{E}[\mathcal{R}_T^\lambda] \leq 11(\lambda K)^{1/3}T^{2/3}.$$

Indeed, using the above observation when $T/B$ may not be a natural number,

$$\mathbb{E}[\mathcal{R}_T^\lambda] \leq B\left(4\sqrt{K(T/B+1)} + 1\right) + \lambda\left(T/B + 1\right) \qquad \text{(Eq. (6))}$$
$$\leq 7\sqrt{KTB} + \lambda\left(2T/B\right)$$
$$\leq 11(\lambda K)^{1/3}T^{2/3}. \qquad (B := \lceil \lambda^{2/3}K^{-1/3}T^{1/3}\rceil)$$

## 5.2 Proof of Theorem 1

"Switch Tsallis, Switch!" consists of two parts. We will denote the regret and pseudo-regret attained after the first part by $\mathcal{R}^{(1)}$ and $\overline{\mathcal{R}}^{(1)}$ respectively, while the regret and pseudo-regret achieved in Line 9 are denoted by $\mathcal{R}^{(2)}$ and $\overline{\mathcal{R}}^{(2)}$. Similarly, $\mathcal{S}^{(1)}$ and $\mathcal{S}^{(2)}$ express the number of switches of each segment. The proof of the adversarial case, i.e. Eq. (4) is straightforward and follows by explicitly bounding the regret in each of these phases:

$$
\begin{aligned}
\mathbb{E}\left[\mathcal{R}_T^\lambda\right] &\leq \mathbb{E}\left[\mathcal{R}^{(1)} + \lambda\mathcal{S}^{(1)}\right] + \mathbb{E}\left[\mathcal{R}^{(2)} + \lambda\mathcal{S}^{(2)}\right] \\
&\leq \mathbb{E}\left[\mathcal{R}^{(1)} + \lambda\mathcal{S}^{(1)}\right] + 12(\lambda K)^{1/3}T^{2/3} && \text{(Corollary 4)} \\
&\leq \mathbb{E}\left[\mathcal{R}^{(1)}\right] + (\lambda K)^{1/3}T^{2/3} + \lambda + 12(\lambda K)^{1/3}T^{2/3} \\
&\leq 4\sqrt{KT} + 1 + \lambda + 13(\lambda K)^{1/3}T^{2/3} && \text{(Eq. (6))} \\
&\leq \mathcal{O}\left((\lambda K)^{1/3}T^{2/3}\right). && (\lambda \geq \sqrt{K/T})
\end{aligned}
$$

We thus continue to the stochastically constrained case. For the proof we rely on the following two Lemmas:

**Lemma 5.** *For every loss sequence, Algorithm 1 satisfies:*

$$
\mathbb{E}\left[\overline{\mathcal{R}}^{(2)} + \lambda\mathcal{S}^{(2)}\right] \leq 11\lambda\,\mathbb{E}\left[\mathcal{S}^{(1)}\right].
$$

**Proof.** Consider the switching cost regret of Line 9 (Algorithm 2) conditioned on $\mathcal{S}^{(1)}$,

$$
\begin{aligned}
\mathbb{E}\left[\overline{\mathcal{R}}^{(2)} + \lambda\mathcal{S}^{(2)}\big|\mathcal{S}^{(1)}\right] &\leq \mathbb{E}\left[\mathcal{R}^{(2)} + \lambda\mathcal{S}^{(2)}\big|\mathcal{S}^{(1)}\right] \\
&\leq \begin{cases} 11(\lambda K)^{1/3}T^{2/3}, & \text{if } \mathcal{S}^{(1)} \geq K^{1/3}(\frac{T}{\lambda})^{2/3} \\ 0, & \text{o.w.} \end{cases} && \text{(Corollary 4)} \\
&\leq \begin{cases} 11(\lambda K)^{1/3}T^{2/3}, & \text{if } \mathcal{S}^{(1)} \geq K^{1/3}(\frac{T}{\lambda})^{2/3} \\ 11\lambda\mathcal{S}^{(1)}, & \text{o.w.} \end{cases} \\
&\leq 11\lambda\mathcal{S}^{(1)}. && (8)
\end{aligned}
$$

Where we used that by the general regret definition we have that for any algorithm $\mathbb{E}\left[\overline{\mathcal{R}}_T\right] \leq \mathbb{E}\left[\mathcal{R}_T\right]$. Taking expectation on both sides of Eq. (8) concludes the proof. ∎

**Lemma 6.** *Suppose we run Algorithm 1 against a stochastically constrained loss sequence with gap $\Delta_{min}$, then:*

$$
\lambda\,\mathbb{E}[\mathcal{S}^{(1)}] \leq \min\left\{\lambda + 2\lambda\frac{\mathbb{E}[\overline{\mathcal{R}}^{(1)}]}{\Delta_{min}}, \lambda + (\lambda K)^{1/3}T^{2/3}\right\}.
$$

**Proof.** Consider some arbitrary arm $i \in [K]$. When a switch occurs at round $t$, either $I_{t-1}$ or $I_t$ are different from $i^\star$. Hence, using linearity of expectation one can bound the expected number of switches $\mathbb{E}[\mathcal{S}]$, regardless to the environment regime (either adversarial or stochastically-constrained adversarial) as follows:

$$
\mathbb{E}[\mathcal{S}] \leq 1 + 2\sum_{t \in [T]}\sum_{i \neq i^\star}\mathbb{E}[p_{t,i}],
$$

where we have used the fact that $\mathbb{P}(I_t = i) = \mathbb{E}[p_{t,i}]$. Additionally, in the stochastically-constrained regime we also have that:

$$
\sum_{t \in [T]}\sum_{i \neq i^\star}\mathbb{E}[p_{t,i}] \leq \sum_{t \in [T]}\sum_{i \neq i^\star}\frac{\mathbb{E}[p_{t,i}]\Delta_i}{\Delta_{\min}} = \frac{\mathbb{E}[\overline{\mathcal{R}}^{(1)}]}{\Delta_{\min}}.
$$

Utilizing the stopping criterion of "Switch Tsallis, Switch!," we can bound the expected number of switches $\mathbb{E}[\mathcal{S}^{(1)}]$, regardless to the environment regime (either adversarial or stochastically-constrained adversarial) and obtain the desired result ∎

The total expected pseudo-regret is upper bounded by the summation of the pseudo-regret attained for each part of the algorithm. We will consider two cases.

$$
\begin{aligned}
\mathbb{E}\big[\overline{\mathcal{R}}_T^{\lambda}\big] &\leq \mathbb{E}\big[\overline{\mathcal{R}}^{(1)} + \lambda \mathcal{S}^{(1)} + \overline{\mathcal{R}}^{(2)} + \lambda \mathcal{S}^{(2)}\big] \\
&\leq \mathbb{E}\big[\overline{\mathcal{R}}^{(1)}\big] + 12\lambda \, \mathbb{E}\big[\mathcal{S}^{(1)}\big] && \text{(Lemma 5)} \\
&\leq \mathbb{E}\big[\overline{\mathcal{R}}^{(1)}\big] + \min\left\{12\lambda + \frac{24\lambda}{\Delta_{\min}} \, \mathbb{E}\big[\overline{\mathcal{R}}^{(1)}\big], 12\lambda + 12(\lambda K)^{1/3} T^{2/3}\right\} && \text{(Lemma 6)} \\
&\leq \min\left\{12\lambda + \Big(\frac{24\lambda}{\Delta_{\min}} + 1\Big) \mathbb{E}\big[\overline{\mathcal{R}}^{(1)}\big], 12\lambda + 12(\lambda K)^{1/3} T^{2/3} + \mathbb{E}\big[\overline{\mathcal{R}}^{(1)}\big]\right\} \\
&\leq \mathcal{O}\left(\min\left\{\Big(\frac{\lambda}{\Delta_{\min}} + 1\Big) \mathbb{E}\big[\overline{\mathcal{R}}^{(1)}\big], (\lambda K)^{1/3} T^{2/3} + \mathbb{E}\big[\overline{\mathcal{R}}^{(1)}\big]\right\}\right) \\
&\leq \mathcal{O}\left(\min\left\{\Big(\frac{\lambda}{\Delta_{\min}} + 1\Big) \sum_{i \neq i^\star} \frac{\log T}{\Delta_i}, (\lambda K)^{1/3} T^{2/3} + \sqrt{KT}\right\}\right). && \text{(Eqs. (6) and (7))}
\end{aligned}
$$

### 5.3 Proof of Theorem 2

Our lower bound builds upon the work of Dekel et al. [8] and we adapt it to the stochastically-constrained adversarial regime. Dekel et al. [8] suggested the following process, depicted in Algorithm 3, to generate an adversarial loss sequence. With correct choice of parameter $\Delta = O(T^{-1/3})$, the process ensures for any deterministic player, a regret of order $\mathbb{E}[\mathcal{R}_T] = \tilde{\Omega}(K^{1/3} T^{2/3})$. For our purposes we need to take care of two things: First we need to generalize the bound to arbitrary $\Delta$. Second, one can see that the loss sequence generated by the adversary in Algorithm 3, is not stochastically constrained (as defined in Section 2). To cope with this, we develop a more fine-grained analysis over a modified loss sequence that assures the stochastically-constrained assumption is met. Towards proving Theorem 2, we present the following Lemmas.

**Lemma 7.** *Let* $\{\ell_1, \ldots, \ell_T\}$ *be the stochastic sequence of loss functions defined in Algorithm 3 for* $K = 2$. *Then for* $T \geq 4$ *and any deterministic player against this sequence it holds*

$$
\mathbb{E}[\mathcal{R}_T + \mathcal{S}_T] \geq \min\{1/(40^2 \Delta^2 \log_2^3 T), \Delta T/24\}.
$$

**Lemma 8.** *Let* $\{\ell_1, \ldots, \ell_T\}$ *be the stochastic sequence of loss functions defined in Algorithm 3 with* $\Delta \leq a K^{1/3} T^{-1/3} \log_2^{-9/2} T$. *Then for* $T \geq \tau$ *and any deterministic player against this sequence, with a switching cost regret guarantee of* $\mathcal{O}(K^{1/3} T^{2/3})$, *against an arbitrary sequence, it holds*

$$
\mathbb{E}[\mathcal{R}_T + \mathcal{S}_T] \geq c K^{1/3} T^{2/3}/\log_2^3 T,
$$

*for some universal constants* $a, c, \tau > 0$.

We deter the proofs of Lemmas 7 and 8 to the supplementary material, and we now proceed with proving our desired lower bound in Theorem 2.

**Proof of Theorem 2.** It can be observed, that the process depicted in Algorithm 3 is almost stochastically constrained, in fact, if at $t$ and $i$ we do not perform clipping to $[0,1]$, i.e. $\ell_t(i) = \ell'_t(i)$ the sequence is indeed stochastically constrained. Let us define then an event $H$ as follows,

$$
H = \left\{\forall t \in [T] : \ X_t + \tfrac{1}{2} \in \big[\tfrac{1}{6}, \tfrac{5}{6}\big]\right\}.
$$

If we restrict $\Delta \leq \frac{1}{6}$, the event $H$ implies that $\ell_t = \ell'_t$ for all $t \in [T]$. By standard Gaussian arguments one can derive the following Lemma.

**Lemma 9** ([8, Lemma 1]). *For any* $\delta \in (0,1)$ *it holds*

$$
\mathbb{P}\big(\big\{\forall t \in [T] : |X_t| \leq 2\sigma \sqrt{\log_2 T \log(T/\delta)}\big\}\big) \geq 1 - \delta.
$$

Setting $\delta = 1/T$ and $\sigma = 1/(9 \log_2 T)$ we get,

$$
\mathbb{P}(H) = \mathbb{P}\big(\big\{\forall t \in [T] : |X_t| \leq \tfrac{1}{3}\big\}\big) \geq 1 - 1/T.
$$

---

**Algorithm 3** The adversary's loss generation process proposed by Dekel et al. [8].

    **Input:** time horizon $T$, the minimal gap $\Delta$.
    **Output:** loss sequence $\left\{\ell_t \in [0,1]^K\right\}_{t \in [T]}$.

1: Set $\sigma = (9 \log_2 T)^{-1}$.
2: Draw $T$ independent Gaussian variables - $\{n_t \sim \mathcal{N}(0, \sigma^2)\}_{t \in [T]}$.
3: Define the process $\{X_t\}_{t \in [T]}$ by

$$\forall t \in [T] : X_t = X_{r(t)} + n_t,$$

    where $X_0 = 0$, $r(t) = t - 2^{m(t)}$, and $m(t) = \max\{i \geq 0 : 2^i \text{ divides } t\}$.
4: Choose $i^\star \in [K]$ uniformly at random.
5: For all $t \in [T]$ and $i \in [K]$, set

$$\ell'_t(i) = X_t + \tfrac{1}{2} - \Delta \cdot \mathbb{1}\{i^\star = i\}.$$

    If $0 \leq \ell'_t(i) \leq 1$, set $\ell_t(i) = \ell'_t(i)$, else perform clipping:

$$\ell_t(i) = \min\{\max\{\ell'_t(i), 0\}, 1\}.$$

---

This implies,

$$\mathbb{E}[\mathcal{R}_T + \mathcal{S}_T] \leq \mathbb{E}[\mathcal{R}_T + \mathcal{S}_T | H^c] \cdot \tfrac{1}{T} + \mathbb{E}[\mathcal{R}_T + \mathcal{S}_T | H] \qquad (\mathcal{R}_T + \mathcal{S}_T \geq 0)$$

$$\leq \Delta + 1 + \mathbb{E}[\mathcal{R}_T + \mathcal{S}_T | H] \qquad (\mathcal{R}_T + \mathcal{S}_T \leq (\Delta + 1)T)$$

Taken this together with Lemma 7 we get that for any deterministic player,

$$\mathbb{E}[\mathcal{R}_T + \mathcal{S}_T | H] = \Omega\left(\min\left\{1/(\Delta^2 \log_2^3 T), \Delta T\right\}\right). \qquad (9)$$

Here we used the fact that for any $\Delta > \frac{1}{6}$ there exists a trivial lower bound of 1. Clearly, a simple derivation shows that the lower bound in Eq. (9) holds for any $K \geq 2$, as we can always extend the loss sequence for $K > 2$ by setting $\ell_{t,i} = 1$ for any $i > 2$. In addition, using Lemma 8, when $\Delta \leq \mathcal{O}(K^{1/3} T^{-1/3} \log_2^{-9/2} T)$, we get that for any deterministic player with a guarantee of $\mathcal{O}(K^{1/3} T^{2/3})$ switching regret,

$$\mathbb{E}[\mathcal{R}_T + \mathcal{S}_T | H] = \Omega\left(K^{1/3} T^{2/3} / \log_2^3 T\right). \qquad (10)$$

Combining both lower bounds in Eqs. (9) and (10) and observing that $\Delta T \geq \Omega(K^{1/3} T^{2/3} / \log_2^{9/2} T)$ when $\Delta \geq \Omega(K^{1/3} T^{-1/3} \log_2^{-9/2} T)$, we obtain that for any $\Delta > 0$,

$$\mathbb{E}[\mathcal{R}_T + \mathcal{S}_T | H] = \Omega\left(\min\left\{1/(\Delta^2 \log_2^3 T), K^{1/3} T^{2/3} / \log_2^{9/2} T\right\}\right). \qquad (11)$$

In other words, if we let the loss sequence to be the conditional process given that $H$ is fulfilled, we obtain a stochastic process that generates a random sequence that satisfies the conditions of a stochastically-constrained loss sequence, i.e. Eq. (1) is met. Eq. (11), in turn, bounds the expected regret given that our loss sequence is drawn from the above process.

Next, since any randomized player can be implemented by a random combination of deterministic players we conclude that Eq. (11) holds for any randomized player. This then immediately implies that there exists a deterministic loss sequence $\{\ell_1, \ldots, \ell_T\}$, that is stochastically-constrained, for which $\mathcal{R}_T + \mathcal{S}_T$ is lower bounded by the RHS of Eq. (11). Lastly, we argue that $\mathbb{E}[\mathcal{R}_T] = \mathbb{E}[\overline{\mathcal{R}}_T]$. This is a direct implication of the loss sequence construction, as for all $t$ we have $i^\star = \arg\min_{i \in [K]} \ell_{t,i}$. Therefore, the acclaimed bounds are achieved also with respect to the pseudo-regret, which concludes the proof. ∎

## 6 Discussion

Best-of-both-worlds algorithm is an extremely challenging setting and in particular, the case of regret with switching cost poses interesting challenges. We presented here an algorithm that achieves (up

to logarithmic factors) optimal minimax rates for the case of two arms, i.e. $K = 2$. Surprisingly, the result is obtained using a very simple modification of the standard best-of-both-worlds Tsallis-INF algorithm. We note that our analysis is agnostic to any best-of-both-worlds algorithm and that Tsallis-INF serves only as a building block of our proposed method. For example, employing the refined bound of Masoudian and Seldin [13] in our analysis follows naturally and will improve our bounds accordingly. Additionally, it is important to mention, that while we assumed the time horizon $T$ is known to the algorithm in advanced, it is not a necessary assumption. One can use a simple doubling trick while preserving the upper bound under the adversarial regime and suffer an extra multiplicative logarithmic factor of $\mathcal{O}(\log T)$ in the stochastically constrained regime so the bound at Eq. (5) becomes,

$$\mathbb{E}[\overline{\mathcal{R}}_T^\lambda] = \mathcal{O}\left(\min\left\{\left(\frac{\lambda \log^2 T}{\Delta_{\min}} + \log^2 T\right)\sum_{i \neq i^\star}\frac{1}{\Delta_i}, (\lambda K)^{1/3}T^{2/3}\right\}\right). \tag{12}$$

Several open problems though seem to arise from our work.

**Open Question 1.** *Given arbitrary K, what is the optimal minimax regret rate, in the stochastically constrained setting, of any algorithm that achieves, in the adversarial regime, regret of*

$$\mathbb{E}[\mathcal{R}_T^\lambda] = \tilde{\mathcal{O}}\left((\lambda K)^{1/3}T^{2/3}\right).$$

In particular, in our result, it is interesting to find out if the term $\mathcal{O}\left(\sum_{i \neq i^\star}\frac{\lambda \log T}{\Delta_{\min}\Delta_i}\right)$ can be replaced by $\mathcal{O}\left(\sum \frac{\lambda \log T}{\Delta_i^2}\right)$. Note that this term is obtained by a worst-case analysis of the switching cost that assumes that we obtained the regret by only switching from the optimal arm to the consecutive second-to-best arm. It seems more likely that any reasonable algorithm roughly switches to each arm $i$, order of $\tilde{\mathcal{O}}(1/\Delta_i^2)$ times, leading to more optimistic rate.

Another natural open problem is to try and generalize the lower bound to the general case. For simplicity we state the next problem for the case all arms have the same gap.

**Open Question 2.** *Suppose $\Delta_1 = \Delta_2 =, \ldots, = \Delta_{min}$, and $\Delta_{min} \geq (\lambda K)^{1/3}T^{-2/3}$. Is it possible to construct an algorithm that achives regret, in the adversarial regime of*

$$\mathbb{E}[\mathcal{R}_T^\lambda] = \tilde{\mathcal{O}}\left((\lambda K)^{1/3}T^{2/3}\right).$$

*and in the stochastically constrained case:*

$$\mathbb{E}[\mathcal{R}_T^\lambda] = o\left(\frac{\lambda K \log T}{\Delta_{min}^2}\right).$$

Finally, we would like to stress that our lower bound applies to a *stochastically constrained* setting, where in principle we often care to understand the stochastic case:

**Open Question 3.** *What is the optimal expected pseudo-regret with switching cost that can be achieved by an algorithm that achieves regret, in the adversarial regime of*

$$\mathbb{E}[\mathcal{R}_T^\lambda] = \tilde{\mathcal{O}}\left((\lambda K)^{1/3}T^{2/3}\right),$$

*against an i.i.d sequence $\ell_1, \ldots, \ell_T$ that satisfies Eq. (1)?*

Achieving a non-trivial lower bound for the above case seems like a very challenging task. In particular, it is known that, if we don't attempt to achieve best-of-both-worlds rate then an upper bound of $O(\sum_{i \neq i^\star}\frac{\log T}{\Delta_i})$ is achievable [9, 10]. Interestingly, then, our lower bound at Theorem 2 presents a separation between the stochastic and stochastically constrained case, leaving open the possibility that a best-of-both-worlds algorithm between adversarial and stochastically constrained case is possible but not necessarily against a stochastic player. Proving the reverse may require new algorithmic techniques. In particular, the current analysis of Tsallis-INF is valid for the stochastically-constrained case as much as to the stochastic case. An improved upper bound for the pure stochastic case, though, cannot improve over the stochastically constrained case as demonstrated by Theorem 2.

**Open Question 4.** *Is the uniqueness of the best arm mandatory in the case of bandits with switching cost?*

The case of multiple best arms introduces new challenges. Whereas Ito [11] showed that Tsallis-INF can handle the case of multiple best arms, it is unclear whether one can use their results to obtain nontrivial bounds in the switching cost setting. In their experiments, Rouyer et al. [14], demonstrated this challenge, suggesting that the requirement of the uniqueness of the best arm is necessary in order to obtain improved bounds. We leave the question of this necessity, in the case of bandits with switching cost and in particular in a best-of-both-worlds setting, to future research.

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
