# Supplementary Material

## A   Proofs of Lemmas 7 and 8

The proofs of Lemmas 7 and 8 require additional notations and some preliminary results. Returning to the process depicted in Algorithm 3, let the conditional probability measures for all $i \in [K]$ be

$$\mathcal{Q}_i(\cdot) = \mathbb{P}(\cdot \mid i^\star = i),$$

and denote $\mathcal{Q}_0$ the probability over the loss sequence when $\Delta = 0$, and all actions incur the same loss. Next, let $\mathcal{F}$ be the $\sigma$-algebra generated by the player's observations $\{\ell_{t,I_t}\}_{t \in [T]}$. Denote the *total variation* distance between $\mathcal{Q}_i$ and $\mathcal{Q}_j$ on $\mathcal{F}$ by

$$d_{\mathrm{TV}}^{\mathcal{F}}(\mathcal{Q}_i, \mathcal{Q}_j) = \sup_{E \in \mathcal{F}} \left| \mathcal{Q}_i(E) - \mathcal{Q}_j(E) \right|.$$

We also denote $\mathbb{E}_{\mathcal{Q}_i}$ as the expectation on the conditional distribution $\mathcal{Q}_i$. Lastly, we present the following result from Dekel et al. [8].

**Lemma 10** ([8, Lemma 3 and Corollary 1]). *For any $i \in [K]$ it holds that*

$$\frac{1}{K} \sum_{i=1}^{K} d_{TV}^{\mathcal{F}}(\mathcal{Q}_0, \mathcal{Q}_i) \leq \frac{\Delta}{\sigma \sqrt{K}} \sqrt{\mathbb{E}_{\mathcal{Q}_0}[\mathcal{S}_T] \log_2 T},$$

*and specifically for $K = 2$,*

$$d_{TV}^{\mathcal{F}}(\mathcal{Q}_1, \mathcal{Q}_2) \leq (\Delta/\sigma) \sqrt{2 \, \mathbb{E}[\mathcal{S}_T] \log_2 T}.$$

With this Lemma at hand, we are ready to prove Lemmas 7 and 8.

**Proof of Lemma 7.** Observe that $\mathcal{R}_T \geq 0$ by the construction in Algorithm 3. Then, if $\mathbb{E}[\mathcal{S}_T] \geq 1/(c\Delta^2 \log_2^3 T)$ for $c = 40^2$ we have that $\mathbb{E}[\mathcal{R}_T + \mathcal{S}_T] \geq 1/(c\Delta^2 \log_2^3 T)$, which guarantees the desired lower bound. On the other hand, applying Lemma 10 when $\mathbb{E}[\mathcal{S}_T] \leq 1/(c\Delta^2 \log_2^3 T)$, we get

$$d_{\mathrm{TV}}^{\mathcal{F}}(\mathcal{Q}_1, \mathcal{Q}_2) \leq (1/\sigma)\sqrt{2/(c \log_2^2 T)} \leq \tfrac{1}{3}. \tag{13}$$

Let $E$ be the event that arm $i = 1$ is picked at least $T/2$ times, namely

$$E = \left\{ \sum_{t \in [T]} \mathbb{1}\{I_t = 1\} \geq T/2 \right\},$$

and let $E^c$ be its complementary event. If $\mathcal{Q}_1(E) \leq \tfrac{1}{2}$ then,

$$
\begin{aligned}
\mathbb{E}[\mathcal{R}_T] &\geq \mathbb{E}_{\mathcal{Q}_1}[\mathcal{R}_T | E^c] \cdot \mathcal{Q}_1(E^c) \cdot \mathbb{P}(i^\star = 1) && (\mathcal{R}_T \geq 0) \\
&\geq \Delta T/8. && (\mathcal{R}_T \geq \Delta T/2 \text{ under the conditional event})
\end{aligned}
$$

If $\mathcal{Q}_1(E) > \tfrac{1}{2}$ then from Eq. (13) we obtain that $\mathcal{Q}_2(E) \geq \tfrac{1}{6}$. This implies,

$$
\begin{aligned}
\mathbb{E}[\mathcal{R}_T] &\geq \mathbb{E}_{\mathcal{Q}_2}[\mathcal{R}_T | E] \cdot \mathcal{Q}_2(E) \cdot \mathbb{P}(i^\star = 2) && (\mathcal{R}_T \geq 0) \\
&\geq \Delta T/24. && (\mathcal{R}_T \geq \Delta T/2 \text{ under the conditional event})
\end{aligned}
$$

Since $\mathcal{S}_T \geq 0$ we conclude the proof. ∎

**Proof of Lemma 8.** The proof is comprised of two steps. First, we prove the lower bound for deterministic players that make at most $K^{1/3} T^{2/3}$ switches. Towards the end of the proof we generalize our claim to any deterministic player. To prove the former, we present the next Lemma, which follows from the proof in [8, Thm 2]. For completeness the proof for this Lemma is provided at the end of the section.

**Lemma 11.** *For any deterministic player that makes at most $\Delta T$ switches over the sequence defined in Algorithm 3,*

$$\mathbb{E}[\mathcal{R}_T + \mathcal{S}_T] \geq \tfrac{1}{3}\Delta T + \mathbb{E}_{\mathcal{Q}_0}[\mathcal{S}_T] - \frac{18\Delta^2 T}{\sqrt{K}} \log_2^{3/2} T \sqrt{\mathbb{E}_{\mathcal{Q}_0}[\mathcal{S}_T]},$$

*provided that $\Delta \leq 1/6$ and $T > 6$.*

Setting $\Delta = \frac{1}{6}$ in Lemma 11 we get,

$$\mathbb{E}[\mathcal{R}_T + \mathcal{S}_T] \geq \frac{1}{18}T + \mathbb{E}_{\mathcal{Q}_0}[\mathcal{S}_T] - \frac{T \log_2^{3/2} T}{2\sqrt{K}} \sqrt{\mathbb{E}_{\mathcal{Q}_0}[\mathcal{S}_T]} \tag{14}$$

In addition, recall that we are interested in deterministic players that satisfy the following regret guarantee in the adversarial regime,

$$\mathbb{E}[\mathcal{R}_T + \mathcal{S}_T] \leq \mathcal{O}(K^{1/3}T^{2/3}). \tag{15}$$

Hence, taking Eqs. (14) and (15) we have,

$$\mathcal{O}(K^{1/3}T^{2/3}) \geq \frac{1}{18}T + \mathbb{E}_{\mathcal{Q}_0}[\mathcal{S}_T] - \frac{T \log_2^{3/2} T}{2\sqrt{K}} \sqrt{\mathbb{E}_{\mathcal{Q}_0}[\mathcal{S}_T]} \tag{16}$$

Now, assuming that $\sqrt{\mathbb{E}_{\mathcal{Q}_0}[\mathcal{S}_T]} < \frac{\sqrt{K}}{10 \log_2^{3/2} T}$ we get that for every $K < T$:

$$\mathcal{O}(K^{1/3}T^{2/3}) \geq \frac{1}{18}T + \mathbb{E}_{\mathcal{Q}_0}[\mathcal{S}_T] - \frac{T \log_2^{3/2} T}{2\sqrt{K}} \sqrt{\mathbb{E}_{\mathcal{Q}_0}[\mathcal{S}_T]}$$
$$> \frac{T}{18} - \frac{T}{20} = \Omega(T)$$

Which is a contradiction. Therefore, in our case, $\sqrt{\mathbb{E}_{\mathcal{Q}_0}[\mathcal{S}_T]} \geq \frac{\sqrt{K}}{10 \log_2^{3/2} T}$. Furthermore, Lemma 11 also holds for any deterministic player that makes at most $K^{1/3}T^{2/3}$ switches, which is less than $\Delta T$ under the condition that $\Delta \geq K^{1/3}T^{-1/3}$. Suppose that $\mathbb{E}_{\mathcal{Q}_0}[\mathcal{S}_T] \leq K^{1/3}T^{2/3}/(60^2 \log_2^3 T)$, then choosing $\frac{1}{6} \geq \Delta = \sqrt{K}/(60\sqrt{\mathbb{E}_{\mathcal{Q}_0}[\mathcal{S}_T]} \log_2^{3/2} T) \geq K^{1/3}T^{-1/3}$ we obtain,

$$\mathbb{E}[\mathcal{R}_T + \mathcal{S}_T] \geq \mathbb{E}_{\mathcal{Q}_0}[\mathcal{S}_T] + \frac{\sqrt{K}T}{3 \cdot 10^3 \sqrt{\mathbb{E}_{\mathcal{Q}_0}[\mathcal{S}_T]} \log_2^{3/2} T}. \tag{17}$$

Taking both observations in Eqs. (15) and (17) implies that $\mathbb{E}_{\mathcal{Q}_0}[\mathcal{S}_T] \geq \Omega(K^{1/3}T^{2/3}/\log_2^3 T)$. To put simply, we have shown that for any deterministic player that makes at most $K^{1/3}T^{2/3}$ switches and holds Eq. (15), then

$$\mathbb{E}_{\mathcal{Q}_0}[\mathcal{S}_T] \geq \Omega(K^{1/3}T^{2/3}/\log_2^3 T), \tag{18}$$

independently of $\Delta$. On the other hand, for any $\Delta > 0$, since $\mathcal{Q}_i(\mathcal{S}_T > K^{1/3}T^{2/3}) = 0$ for any $i \in [K] \cup \{0\}$,

$$\mathbb{E}_{\mathcal{Q}_0}[\mathcal{S}_T] - \mathbb{E}_{\mathcal{Q}_i}[\mathcal{S}_T] = \sum_{s=1}^{\lfloor K^{1/3}T^{2/3} \rfloor} (\mathcal{Q}_0(\mathcal{S}_T \geq s) - \mathcal{Q}_i(\mathcal{S}_T \geq s))$$
$$\leq K^{1/3}T^{2/3} \cdot d_{\text{TV}}^{\mathcal{F}}(\mathcal{Q}_0, \mathcal{Q}_i).$$

Averaging over $i$ and rearranging terms we get,

$$\mathbb{E}[\mathcal{S}_T] \geq \mathbb{E}_{\mathcal{Q}_0}[\mathcal{S}_T] - \frac{T^{2/3}}{K^{2/3}} \sum_{i=1}^{K} d_{\text{TV}}^{\mathcal{F}}(\mathcal{Q}_0, \mathcal{Q}_i)$$
$$\geq \mathbb{E}_{\mathcal{Q}_0}[\mathcal{S}_T] - 9\Delta K^{-1/6}T^{2/3} \log_2^{3/2} T \sqrt{\mathbb{E}_{\mathcal{Q}_0}[\mathcal{S}_T]} \tag{Lemma 10}$$

Using Eq. (18) and the assumption $\mathcal{S}_T \leq K^{1/3}T^{2/3}$, we get that for any $\Delta \leq aK^{1/3}T^{-1/3} \log_2^{-9/2} T$ for some constant $a > 0$ and sufficiently large $T$,

$$\mathbb{E}[\mathcal{R}_T + \mathcal{S}_T] \geq \mathbb{E}[\mathcal{S}_T] \geq \Omega(K^{1/3}T^{2/3}/\log_2^3 T). \tag{19}$$

The above lower bound holds for any deterministic player that makes at most $K^{1/3}T^{2/3}$ switches. However, given a general deterministic player denoted by $A$ we can construct an alternative player,

denoted by $\tilde{A}$, which is identical to $A$, up to the round $A$ performs the $\lfloor \frac{1}{2}K^{1/3}T^{2/3}\rfloor$ switch. After that $\tilde{A}$ employs the Tsalis-INF algorithm with blocks of size $B = \lceil 4K^{-1/3}T^{1/3}\rceil$ for the remaining rounds (see Algorithm 2). Clearly, the number of switches this block algorithm does is upper bounded by $T/B + 1 \le K^{1/3}T^{2/3}/2$, therefore $\tilde{A}$ performs at most $K^{1/3}T^{2/3}$ switches. We denote, $\mathcal{R}_T^A + \mathcal{S}_T^A$ the regret with switching cost of player $A$ and $\mathcal{R}_T^{\tilde{A}} + \mathcal{S}_T^{\tilde{A}}$ respectively. Observe that when $\mathcal{S}_T^A < \lfloor \frac{1}{2}K^{1/3}T^{2/3}\rfloor$ we get,

$$\mathcal{R}_T^A + \mathcal{S}_T^A = \mathcal{R}_T^{\tilde{A}} + \mathcal{S}_T^{\tilde{A}}.$$

While for $\mathcal{S}_T^A \ge \lfloor \frac{1}{2}K^{1/3}T^{2/3}\rfloor$,

$$\mathcal{R}_T^{\tilde{A}} + \mathcal{S}_T^{\tilde{A}} \le \mathcal{R}_T^A + \mathcal{S}_T^A + 21K^{1/3}T^{2/3} \qquad \text{(Corollary 4 with } B = \lceil 4K^{-1/3}T^{1/3}\rceil)$$
$$\le \mathcal{R}_T^A + 63\mathcal{S}_T^A. \qquad (\mathcal{S}_T^A \ge \tfrac{1}{3}K^{1/3}T^{2/3} \text{ for } T \ge 15)$$

This implies that $\mathcal{R}_T^A + \mathcal{S}_T^A \ge \frac{1}{63}(\mathcal{R}_T^{\tilde{A}} + \mathcal{S}_T^{\tilde{A}})$, and together with Eq. (19) it concludes the proof. ∎

**Proof of Lemma 11.** We examine deterministic players that make at most $\Delta T$ switches. Since $S_T \le \Delta T$ we have that,

$$\mathbb{E}_{\mathcal{Q}_0}[\mathcal{S}_T] - \mathbb{E}_{\mathcal{Q}_i}[\mathcal{S}_T] = \sum_{s=1}^{\lceil \Delta T\rceil}(\mathcal{Q}_0(\mathcal{S}_T \ge s) - \mathcal{Q}_i(\mathcal{S}_T \ge s)) \qquad (\mathcal{Q}_i(\mathcal{S}_T > \Delta T) = 0 \ \forall i \in [K] \cup \{0\})$$
$$\le \Delta T \cdot d_{\text{TV}}^{\mathcal{F}}(\mathcal{Q}_0, \mathcal{Q}_i).$$

Averaging over $i$ and rearranging terms we get,

$$\mathbb{E}[\mathcal{S}_T] \ge \mathbb{E}_{\mathcal{Q}_0}[\mathcal{S}_T] - \frac{\Delta T}{K}\sum_{i=1}^{K} d_{\text{TV}}^{\mathcal{F}}(\mathcal{Q}_0, \mathcal{Q}_i). \qquad (20)$$

Next we present the following Lemma that is taken verbatim from Dekel et al. [8].

**Lemma 12** ([8, Lemmas 4 and 5]). *Assume that $T \ge \max\{K, 6\}$ and $\Delta \le 1/6$ then,*

$$\mathbb{E}[\mathcal{R}_T + \mathcal{S}_T] \ge \frac{\Delta T}{3} - \frac{\Delta T}{K}\sum_{i=1}^{K} d_{TV}^{\mathcal{F}}(\mathcal{Q}_0, \mathcal{Q}_i) + \mathbb{E}[\mathcal{S}_T].$$

Using Lemma 12 together with Eq. (20) we obtain,

$$\mathbb{E}[\mathcal{R}_T + \mathcal{S}_T] \ge \frac{\Delta T}{3} - \frac{2\Delta T}{K}\sum_{i=1}^{K} d_{\text{TV}}^{\mathcal{F}}(\mathcal{Q}_0, \mathcal{Q}_i) + \mathbb{E}_{\mathcal{Q}_0}[\mathcal{S}_T]$$
$$\ge \frac{\Delta T}{3} - \frac{2\Delta^2 T}{\sigma\sqrt{K}}\sqrt{\mathbb{E}_{\mathcal{Q}_0}[\mathcal{S}_T]\log_2 T} + \mathbb{E}_{\mathcal{Q}_0}[\mathcal{S}_T] \qquad \text{(Lemma 10)}$$
$$= \frac{\Delta T}{3} - \frac{18\Delta^2 T}{\sqrt{K}}\log_2^{3/2}T\sqrt{\mathbb{E}_{\mathcal{Q}_0}[\mathcal{S}_T]} + \mathbb{E}_{\mathcal{Q}_0}[\mathcal{S}_T]. \qquad (\sigma = 1/(9\log_2 T))$$

Setting $\sigma = 1/(9\log_2 T)$ we conclude,

$$\mathbb{E}[\mathcal{R}_T + \mathcal{S}_T] \ge \frac{\Delta T}{3} - \frac{18\Delta^2 T}{\sqrt{K}}\log_2^{3/2}T\sqrt{\mathbb{E}_{\mathcal{Q}_0}[\mathcal{S}_T]} + \mathbb{E}_{\mathcal{Q}_0}[\mathcal{S}_T].$$

∎