# OpenReview forum: "Better Best of Both Worlds Bounds for Bandits with Switching Costs"
_NeurIPS.cc/2022/Conference — NeurIPS 2022 Accept_

### Official Review · Reviewer_qa8x · 2022-06-16

**Rating:** 7
**Confidence:** 5
**Soundness:** 4 excellent
**Presentation:** 3 good
**Contribution:** 3 good

**Summary:**

The paper considers the problem of bandits with switching costs and studies it in a best-of-both worlds setting. Bandits with switching costs have been extensively studied in both the adversarial and the stochastic regimes separately, but only recently considered in the the best-of-both worlds setting by Rouyer, Seldin and Cesa-Bianchi (2021).

The authors consider the same regime as Rouyer et al. and present a simple algorithm that achieves the same (minmax optimal) rate of $O((\lambda K)^{1/3}T^{2/3})$ pseudo-regret against oblivious adversarial sequences of losses, and a $polylog(T)/\Delta^2$ pseudo-regret against stochastically constrained adversarial sequences of losses. This is the first polylog type of bound for this regime.
The algorithm that they propose uses the regular Tsallis-Inf algorithm until the cumulative switching cost becomes too large, which triggers a change of regime to a version of Tsallis-Inf played with blocks. This type of algorithms was initially studied by Auer and Chiang (2016) in order to design a a near optimal best-of-both worlds algorithm. In this problem, as the algorithm is aware of the switching costs, monitoring when to trake the change does not incur extra logarithmic factors because the algorithm can use the switching cost to know when to trigger the change of regime rahter than by having to estimate the regret.

Furthermore, the authors derive a lower bound for the stochastically-constrained regime in $\tilde \Omega (\min(1/\Delta^2, \Delta T)) $, which shows that the dependency in terms of $1/\Delta^2$ is unavoidable, and highlights that for this problem, the stochastically constrained regime is more difficult than the stochastic regime (which is a special case of stochastically constraine adversarial regime).

They finish off by mentioning several open problems that highlight the difficult aspects of the problem.  Those include whether it is possible to improve the dependency on the suboptimality gaps when $K > 2$ (by a factor $\Delta_{min} / \Delta_i$ on each arm), whether is is possible to achieve some improved result when all the suboptimality gaps are equal and larger than $1/T^{2/3}$, and question what the lower bound for the purely stochastic regime would be in a best-of-both worlds setting.


**Questions:**

Have you thought about the points discussed in the previous section? Namely about the unicity of the best arm and the required knowledge of the time horizon?



**Limitations:**

Theoretical work, so societal impact is N/A.

The paper is clear with the assumptions that are made for their results to apply, and the capacity to generalize those assumptions has already been mentionned earlier.



**Strengths And Weaknesses:**

The main contribution if the paper is certainly the lower bound that they derived.  Showing that there is a difference between the lower bounds for the stochastic and the stochastically constrained adversarial regimes is a crucial step towards understanding best-of-all worlds types of results, not only for bandits with switching costs but possibly for other variations of bandits problems.

The lower bound proposed is not tight up to logarithmic factors. That bound generalizes the analysis in the adversarial regime (Dekel et al. (2014)) which also contained logarithmic factors.

The algorithm proposed is simple and surprisingly well suited for the task. Algorithms that start by assuming that they are in a certain regime and then switch to another regime in case those assumptions are broken have been analysed when looking at best-of-both worlds bandits (see Auer et Chiang 2016), but having access to the switching costs makes the analysis easier, thus not leading to any extra logarithmic factors.

This being said, one of the limitations of the algorithm is that it may not be well suited to generalizations of the problem.
In this paper, it is assumed that the best arm is unique. This assumption is not trivial in the case of bandits with switching costs, because in presence of several optimal arms, the algorithm may often switch between those optimal arms often, which may lead to a a large cumulative switching cost, and thus would force the algorithm to switch regimes early, leading to a $O(T^{2/3})$ type of regret. [Note that while it was originally unclear whether Tsallis-Inf could handle several optimal arms, it has been shown recently (Ito (2021)) that Tsallis-Inf can handle several optimal arms]
Also, the algorithm requires the time horizon to be known by the learner. It is likely that some type of doubling trick could be used to estimate the time horizon, but choosing the rate at which the time horizon increases may lead to logarithmic factors in at least one of the regimes.

The paper is clearly written and compact.

---

> ### Author Response · Authors · 2022-08-02
> **Thanks for supporting acceptance**
>
> > ”Also, the algorithm requires the time horizon to be known by the learner.”
>
> This is a good point and we will discuss it more at length. Please see a more detailed response to **P4Ai** that raises the exact same concern.
>
> > “In this paper, it is assumed that the best arm is unique. This assumption is not trivial in the case of bandits with switching costs …”
>
> We addressed it also in our response to reviewer **P4Ai**, for your convenience we provide our answer here as well.
> This is an interesting question. For us, it is unclear whether one can use the result of Ito (2021) in the case of multiple best arms to obtain non-trivial bounds in the switching cost setting. Multiple best arms in the switching cost setting seems to introduce new challenges. The only indicator we are aware of is the one by Rouyer et al. (2021), in which their empirical experiments suggest that the requirement of the uniqueness of the best arm is necessary in order to obtain improved bounds.

---

> > ### Comment · Reviewer_qa8x · 2022-08-07
> > **thanks for the clarifications**
> >
> > Hi,
> > Thank you for the clarifications. I believe that because handling the time horizon with a doubling trick provides an extra logarithmic factor, it is important to mention this in the paper.
> >
> > Best

---

### Official Review · Reviewer_5PVD · 2022-06-22

**Rating:** 6
**Confidence:** 4
**Soundness:** 3 good
**Presentation:** 3 good
**Contribution:** 3 good

**Summary:**

This paper considers the MAB problem with switching costs. The authors try to design a best-of-both-worlds algorithm, i.e., its regret upper bound is optimal for both the stochastic case and the adversarial case. In this paper, instead of the stochastic case, the authors consider a stochastically-constrained adversarial setting, in which the adversary needs to choose the losses from random distributions such that the expected loss of the optimal arm is always smaller than the sub-optimal arms by at least $\Delta$ at each time step. For this problem setting, the authors design an algorithm that achieves $O(\lambda^{1/3}K^{1/3}T^{2/3})$ regret upper bound in the adversarial case, where $\lambda$ is the switching cost, $K$ is the number of arms and $T$ is the time horizon. Existing results show that this is the best one can do in the adversarial case. As for the stochastically-constrained case, the algorithm can achieve a regret upper bound of $O({K\lambda \log T \over \Delta^2})$, and the authors also show that this regret upper bound is optimal (up to logarithmic factors) in the stochastically-constrained case by proving a matching regret lower bound.

**Questions:**

1. It is mentioned that in the stochastic case, [9,10] can achieve a regret upper bound of $O({K\log T \over \Delta})$. However, Theorem 2 shows that the regret lower bound for the stochastically-constrained case is $\Omega({K \log T \over \Delta^2})$. My question is that, in high-level ideas, what is the reason for this difference? Why is the i.i.d. assumption important to reduce the regret of bandits with switching costs?

2. This paper only considers the oblivious adversary. What if the adversary is non-oblivious?

**Limitations:**

Yes.

**Strengths And Weaknesses:**

1. About the significance.

The best-of-both-worlds algorithms in the bandit setting attract much attention in recent years. The bandit problem with switching costs is also a well-motivated model. I also really like the natural idea of improving Tsallis-Switch by adding a start phase that runs classic Tsallis-INF and only changes to Tsallis-Switch if there are too many switches before. My main concern here is that "Switch Tsallis, Switch!" is not a fully best-of-both-world algorithm, i.e., it is best under the stochastically-constrained case but not the stochastic case.

2. About the clarity.

Overall, the presentation is good and clear, though there are some typos that need to fix, e.g.,

- i) Line 51: "Notice, that" to "Notice that".

- ii) Line 83: "Where" to "Here".

- iii) Equation (3): missing a period in the end.

3. About the proofs.

I check all the proofs and they seem to be correct.

4. Some experimental results of comparison between "Tsallis-Switch" and "Switch Tsallis, Switch!" (for both the stochastic case and the adversarial case) would be helpful.

---

> ### Author Response · Authors · 2022-08-02
> **Thank you for the review**
>
> > “My main concern here is that "Switch Tsallis, Switch!" is not a fully best-of-both-world algorithm, i.e., it is best under the stochastically-constrained case but not the stochastic case. “
>
> You are indeed correct, and we do discuss this issue in the paper (see open question 3).  We agree that obtaining a best-of-both world algorithm for stochastic and adversarial setting is a worthy goal.  Nonetheless, to our knowledge, prior to our work there was no stated separation between the stochastic and the stochastically-constrained regimes in this context.  So in that sense (as **qa8x** points out), we are making a crucial step towards understanding best-of-all worlds types of results for bandit problems.
>
> > Minor typos
>
> Thanks, we will fix those.
>
>
>
> > “It is mentioned that in the stochastic case, [9,10] can achieve a regret upper bound of $\mathcal{O}(K\log⁡ T/\Delta)$. However, Theorem 2 shows that the regret lower bound for the stochastically-constrained case is $\Omega(K\log ⁡T / \Delta^2)$. My question is that, in high-level ideas, what is the reason for this difference? Why is the i.i.d. assumption important to reduce the regret of bandits with switching costs?”
>
> That is a very interesting question. Please note though, that [9,10] are not in the best-of-both-worlds setup so it is unclear that this upper bound is valid in our setup.
> But to answer the question, very roughly, the construction of the lower bound builds on two non-i.i.d stochastic processes that differ on the initial state (i.e. one arm is constantly better than the other by a constant gap). In this case, switches are necessary as choosing the same arm twice doesn’t provide any new information. Such a construction is impossible in the i.i.d case.
>
> > “This paper only considers the oblivious adversary. What if the adversary is non-oblivious?”
>
> This is a subtle question as in the context of bandits, regret against non-oblivious adversaries can be measured through policy-regret or standard external regret (we use the terminology of [1] here). Notice that switching costs are also regarded as a type of an adaptive adversary, and the relevant regret in this case is the policy-regret (otherwise, the best arm would also inflict switching costs). If the reviewer has a particular setup in mind, we would be very happy to hear about it and address it in more detail during the discussion.
> We do want to point out that the strategy we exploit is quite general and for any setup where Tsallis-INF has regret guarantee $f(T)$, our strategy is to play Tsallis-INF, and change to a worst case regret algorithm with switching cost once $f(T)$ switches are inflicted. This strategy is generalizable for any setup that is harder than the stochastically constrained setup.
>
> [1]: Arora et al. Online Bandit Learning against an Adaptive Adversary: from Regret to Policy Regret

---

### Official Review · Reviewer_DEf3 · 2022-07-03

**Rating:** 6
**Confidence:** 4
**Soundness:** 4 excellent
**Presentation:** 3 good
**Contribution:** 3 good

**Summary:**

This paper considers the problem of achieving best-of-both-worlds in bandits with switching cost. Specifically, the authors design an algorithm that achieves O(T^{2/3}) regret bound when the loss environment is adversarial and $\min\{O(\frac{\log(T)}{\Delta_{\min}} (\sum_i \frac{1}{\Delta_i})), T^{2/3}\}$ regret bound when the environment is stochastically-constrained. The algorithm is based on a combination of classic FTRL with Tsallis-entropy regularizer and the block-wise update Tsallis-INF, which intuitively makes sense. The authors also show a lower bound of $\Omega(1/\Delta^2, \Delta T)$ in this problem when K=2.

**Questions:**

Could the authors explain more about the novelty about both the algorithm design and the lower bound proofs as for me, the two issues mentioned in Section 5.3 do not seem to be very difficult to handle based on the original proof shown in [Dekel et al., 2014].

**Limitations:**

Yes, the authors address the limitations of their work.

**Strengths And Weaknesses:**

Strengths:
- The paper is nicely and neatly written.
- The algorithm is intuitive and presented in a very clear way. The analysis is also presented neatly. I check the analysis and they all look correct to me.
- The obtained results improve upon the previous work specifically in the bound for the stochastically-constrained case from $O(T^{1/3}/\Delta)$ to $O(\log(T)/\Delta^2, T^{2/3})$.

Weakness:
- I think there is no major technical issue for this paper. My main concern is the novelty of this work as the proposed algorithm is a simple combination of the known Tsallis-INF algorithm and the block-wise Tsallis-INF algorithm with a hard switch depending on the current number of switches of the algorithm. Lemma 6 looks interesting to me, which relates the number of switch to the regret but the remaining analysis is somewhat direct from the previous analysis. For the lower bound part, I think the modification from [Dekel et al., 2014] is also somewhat minor based on my understanding.

Minor typos:
- line 104: which demonstrate -> which demonstrates
- line 121: quite involve -> quite involved
- line 123: becomes quite -> become quite
- line 155: .Where -> , where
- line 137: I think the block size should be $\lambda^{2/3}K^{-1/3}T^{1/3}$ instead of the one with big O notation as you are using the exact constant in the regret bound.

---

> ### Author Response · Authors · 2022-08-02
> **Thank you for the review**
>
> > “Could the authors explain more about the novelty about both the algorithm design and the lower bound proofs as for me, the two issues mentioned in Section 5.3 do not seem to be very difficult to handle based on the original proof shown in [Dekel et al., 2014].”
>
> The analysis we present is indeed simple. However, we find it as a virtue of our work and it is quite surprising that our simple approach improves on previously known results - $\mathcal{O}(T^{1/3}/\Delta)$ compared to our $\mathcal{O}\big(\min( \log T/\Delta^2, T^{2/3})\big)$.
>
> Specifically, as **P4Ai** points out in her/his review, in general recognizing the type of environment and the time to switch is quite challenging in bandit. Here we identify the fact that, due to switching cost, determining when to switch turns out to be extremely simple.
>
> Regarding the lower bound - see our response to reviewer **P4Ai**.
>
> > Minor typos
>
> Thanks, we will fix those.

---

> > ### Comment · Reviewer_DEf3 · 2022-08-09
> > **Thanks for the response**
> >
> > Thanks for the authors' response and my issues are addressed and I keep my score unchanged.

---

### Official Review · Reviewer_P4Ai · 2022-07-06

**Rating:** 7
**Confidence:** 5
**Ethics Flag:** Yes
**Soundness:** 4 excellent
**Presentation:** 3 good
**Contribution:** 3 good

**Summary:**

The paper provides a simple modification on the best-of-both-worlds algorithm of Rouyer et al.  (2021) to achieve a logarithmic pseudo-regret bound in stochastically-constrained setting which improves $O(T^{1/3}/\Delta)$ bound of Rouyer et al.  (2021) to $O(\min ( \log T/\Delta^2, T^{2/3} ) )$, while achieving the same optimal regret bound for oblivious adversary setting. They also provide a lower bound for stochastically-constrained setting that shows the factor $1/\Delta^2$ is unavoidable.

**Questions:**

As I mentioned in the previous part, I want to see a dicussion on the necessity knowing time horizon in advance. More precisely, I want to know how your results would be affected by removing this knowledge?



**Limitations:**

There are two works where they refined the analysis of Tsallis-INF algorithm:
1. work by Ito (2021), where they remove the uniqueness assumption needed by the analysis of Tsallis-INF.
2. work by Masoudian and Seldin (2021), where they refine the regret bound of Tallis-INF in stochastically contained setting and intermediate regimes.

There is no discussion that what are the limitation that prevents the authors to get the same refinement here since their algorithm is based on Tsallis-INF.



Shinji Ito. Parameter-free multi-armed bandit algorithms with hybrid data-dependent regret bounds. In
Proceedings of Thirty Fourth Conference on Learning Theory, 2021.

Saeed Masoudian and Yevgeny Seldin. Improved analysis of the tsallis-inf algorithm in stochastically constrained
adversarial bandits and stochastic bandits with adversarial corruptions. In Proceedings of Thirty Fourth
Conference on Learning Theory, 2021.

**Strengths And Weaknesses:**

Originality:

- The paper essentially has one general key idea which is to verify and decide when and how to switch to different algorithm, whereas this general idea has been used before to obtain best-of-both-worlds algorithms. While recognizing the type of environment and the time to switch is quite challenging in bandit, here determining when to switch is extremely simple.
While the novelty of provided algorithm is questionable, I believe recognizing this fact that this simple modification is would works here in stochastically-constrained setting is quite interesting.
The analysis of the proof is very straight forward and uses Tsallis-INF bounds along with some bounds on switches which is again not a completely new novel analysis. Therefore the analysis has no special novelty.

- The lower bound part of the paper is basically based on the construction suggested by Dekel et al. (2014), whereas, it originally works for adversary setting and here the authors tried to adapt it to stochastically-constrained setting, as well. I may consider some novelties for the adaptation part but the analysis of it has no new idea and novelties.

Quality and Clarity:

- Generally the flow of paper and the analyses are clear and easy to follow. I tracked all the analysis steps and seems sound to me. However, there are some minor issues listed below.

1. In Algorithm 1, I failed to find definition of learning rates $\eta_t$. I found the definition from Rouyer et al. (2021) but you must also include the definition in your paper as well.
2. There are two typos in the analysis where you use  $S$ instead of $\mathcal{S}$ in proof of Lemma 5 (the second inequality between line 148 and 149) and in the statement of Lemma 6.
3. The sentence that starts from line 121 to 124 is vague to me. I suggest to rephrase it to make it more clear.

Significance:

- Despite the simple trick used by the authors, the results is significant to me from two perspectives. First, they answer to the open problem arised by Rouyer et al. (2021) that asks if it is possible to obtain a logarithmic regret in stochastically-constrained setting with affirmative answer. Second, their lower bound indicates an essential difference between stochastic and stochastically-constrained settings. They show for the later setting the multiplicative factor $1/\Delta^2$ is unavoidable, whereas in a pure stochastic setting the optimal multiplicative factor is $1/\Delta$.

The thorough discussion on the open problems is also considered as a strength of the paper.

The only thing about result that annoys me is that their algorithm requires knowledge of the time horizon in advance, while the previous work that theirs is upon that, needs no prior knowledge of the time horizon. There is no dicussion on the necessity of this assumption and what could be the cost of removing it.

---

> ### Author Response · Authors · 2022-08-02
> **Thank you for the review**
>
> > “The lower bound part of the paper is basically based on the construction suggested by Dekel et al. (2014) … I may consider some novelties for the adaptation part but the analysis of it has no new idea and novelties”
>
> It is true that our bound builds upon previous work, however, as we discuss in the paper, there are two issues to note (1) the analysis of Dekel et al. is tailored towards a specific choice of $\Delta=\mathcal{O}(T^{-1/3})$, and (2) the generated sequence is not really stochastically constrained. Generalizing the construction for any $\Delta$, and making sure the sequence will be stochastically constrained had its technical challenges, and some careful analysis had to be taken.
>
> > “The only thing about the result that annoys me is that their algorithm requires knowledge of the time horizon in advance… There is no discussion on the necessity of this assumption and what could be the cost of removing it.”
>
> Thank you, we truly appreciate the feedback. Indeed, when comparing our bound to previous work this should be part of the discussion and we will discuss this and make the necessary comparison in the final version.  As reviewer **qa8x** suggests, one can modify our algorithm by using the Doubling Trick. In that case, only the stochastically-constrained bound suffers another logarithmic factor resulting in $\mathcal{O}(\log^2 T)$. We are not certain if this can be avoided and it is an interesting question.
>
> > “There are two works where they refined the analysis of Tsallis-INF algorithm:
> > 1. work by Ito (2021), where they remove the uniqueness assumption needed by the analysis of Tsallis-INF.
> > 2. work by Masoudian and Seldin (2021), where they refine the regret bound of Tallis-INF in stochastically contained setting and intermediate regimes.
> >
> > There is no discussion that what are the limitation that prevents the authors to get the same refinement here since their algorithm is based on Tsallis-INF.”
>
> Thank you for bringing these papers to our attention, those are great points that we will add to the main paper.
> 1. This is an interesting question. For us, it is unclear whether one can use the result of Ito (2021) in the case of multiple best arms to obtain non-trivial bounds in the switching cost setting. Multiple best arms in the switching cost setting seems to introduce new challenges. The only indicator we are aware of is the one by Rouyer et al. (2021), in which their empirical experiments suggest that the requirement of the uniqueness of the best arm is necessary in order to obtain improved bounds.
> 2. Indeed, we use Tsalis-INF as a building block of our proposed algorithm. Essentially, our analysis is agnostic to any other algorithm that obtains best-of-both-worlds guarantees. More specifically, employing the refined bound of Masoudian and Seldin (2021) in our analysis follows naturally and will improve our bounds accordingly.
>
>
> > Typos $S$ -> $\mathcal{S}$
>
> Thanks, we will fix those.
>
> > “The sentence that starts from line 121 to 124 is vague to me. I suggest to rephrase it to make it more clear.”
>
> Thanks for pointing that out, we will clarify this sentence. We suggest the following revision:
>
> Moreover, in the standard best-of-both-worlds setup (without switching cost), the optimal methods don’t attempt to identify the regime (stochastic-cosntrained or adversarial). In contrast, what we observe here, is that once switching cost is involved, the criteria to shift between the regimes becomes quite straightforward which allows us to design such a simple algorithm.
>
>
> > “In Algorithm 1, I failed to find definition of learning rates $\eta_t$”
>
> See the initialization part of Algorithm 1.

---

> > ### Comment · Reviewer_P4Ai · 2022-08-09
> > **Response to the Authors**
> >
> > Thanks for addressing my concerns and questions. I gladly increase my score.

---

### Meta-Review · Area_Chair_bgKM · 2022-08-26

**Recommendation:** Accept
**Confidence:** Certain

**Metareview:**

Reviewers all agree that this is an interesting work with significant contribution to the best-of-both-worlds literature. Clear accept. Please do still address the minor issues pointed out by the reviewers in the final version.

**Award:**

No

---

### Decision · Program_Chairs · 2022-09-14

Accept